# The Revolution in Indication for Liver Transplantation: Will Liver Metastatic Disease Overcome the End-Stage Liver Disease in the Next Future?

**Tommaso Maria Manzia \*, Alessandro Parente , Roberta Angelico, Carlo Gazia and Giuseppe Tisone**

Department of Hepatobiliary Surgery and Transplant Unit, Tor Vergata Hospital, Tor Vergata University of Rome, 81, Viale Oxford, 00133 Rome, Italy; aleparen@gmail.com (A.P.); roberta.angelico@uniroma2.it (R.A.); carlogazia9@gmail.com (C.G.); tisone@med.uniroma2.it (G.T.)

**\*** Correspondence: manzia@med.uniroma2.it

**Abstract:** Indications for liver transplantation (LT) have constantly been evolving during the last few decades due to a better understanding of liver diseases and innovative therapies. Likewise, also the underlying causes of liver disease have changed. In the setting of transplant oncology, recent developments have pushed the boundaries of oncological indications for LT outside hepatocellular carcinoma (HCC), especially for secondary liver tumors, such as neuroendocrine and colorectal cancer. In the next years, as more evidence emerges, LT could become the standard treatment for well-selected metastatic liver tumors. In this manuscript, we review and summarize the available evidence for LT in liver tumors beyond HCC with a focus on metastatic liver malignancies, highlighting the importance of these new concepts for future implications.

**Keywords:** liver transplantation; liver metastasis; secondary liver tumors

## 1. Introduction

Liver transplantation (LT) is the only recognized and highly effective treatment for end-stage liver disease (ESLD), acute liver failure (ALF) and well-selected liver cancer. The aims of LT are to improve patients' quality of life and prolong life expectancy. However, organ shortage is worldwide recognized as the main challenge in the transplant community, with a waiting list mortality around 10%. Cirrhosis, namely ESLD, has been the most common indication for LT. In the early 2000s, Model for End-stage Liver Disease (MELD), based on laboratory values of bilirubin, serum creatinine and INR, was developed to establish the extent of the liver disease and it was then approved for the prognosis stratification in patients affected by liver cirrhosis and, subsequently used for organ allocation [1]. The hepatocellular carcinoma (HCC) on cirrhosis is the most common primary cancer of the liver and it is a well-established indication for LT with acceptable outcomes [2]. Other types of primary liver malignancies, such as fibrolamellar carcinoma and epitheliod haemangioendothelioma are also successfully treated with LT. The cholangiocarcinoma is a controversial indication for LT due to the high risk of recurrence despite some protocols that combine neoadjuvant chemotherapy and LT—in perihilar cholangiocarcinoma—achieved a 5-year disease-free survival of 65% [3]. Historically, liver secondary metastatic tumors have been considered a contraindication to LT due to poor outcomes and it was subsequently abandoned [4]. However, recent developments with the new concept of transplant oncology [5] pushed the boundaries of oncological indications for LT, especially for secondary liver cancer. Slowly growing neuroendocrine tumors (NETs) occasionally metastasize to the liver and when

diagnosed the final outcome may be already influenced by their organ diffusion. Nowadays, in selected cases, where surgical resections are not feasible, LT might offer a valid and compelling therapeutic option. The aim of the current review was to summarize the current evidence of LT in secondary liver tumors, focusing on NETs and colorectal liver metastasis (CRLM), in order to understand the growing importance of these new concepts in the field of transplantation.

## 2. Material and Methods: Literature Research

### 2.1. Search Strategy

After accessing full-text eligible studies available in the literature, research was conducted by independent authors through MEDLINE databases (*via* PubMed) to find relevant studies pertaining to LT, colorectal liver metastasis (CRLM) and neuroendocrine tumors (NET). Articles published in languages other than English were excluded, and all selected publications and data were cross-checked and accessed.

Multiple keywords were used: "liver transplantation", "liver metastasis", "colorectal liver metastasis", "neuroendocrine tumor", "liver secondary tumor". The combination of words was used in order to maximize the results and access the highest number of articles related to the field of the present manuscript.

### 2.2. Inclusion and Exclusion Criteria

Studies published in journals describing liver metastatic disease and risk factors for their development were searched for LT recipients, including bibliographies from accessed articles. Records on LT for NETs and CRLM were collected and discussed from randomized clinical trials, systematic reviews, observational and original studies. No time limits were applied to fulfill at best the aim of this review. Non-English articles and cohorts of patients who underwent allografts other than liver were excluded from this manuscript.

## 3. Liver Transplantation for Metastatic Liver Tumours

### 3.1. Neuroendocrine Tumours—Natural History and Approach to Liver Transplantation

NETs are infrequent malignancies arising from the neuroendocrine system. Their behavior varies from patient to patient, since NETs may be indolent or spread with massive metastases and refractory carcinoid syndromes [6]. Primary NETs may develop from different organs, such as small intestine (45%), pancreas (42%), colon (40%), stomach (15%), rectum (6%) and appendix (3%) [7]. Around 5–10% of patients present with liver NETs of unknown primary source.

Pancreatic NETs (PNETs) are the most frequent primary tumors metastasizing to the liver and represents an indication to LT in up to 50% of cases [8,9]. The most common functional PNET is represented by insulinoma, but only less than 10% are malignant, whereas VIPomas and glucagonomas might also be functional tumors even if they represent a smaller amount of PNETs. The European Liver Transplant Registry (ELTR) study, conducted by Le Treut et al., described distal pancreatectomy as the most common surgical procedure to eradicate primary PNETs [7,10]. The 5 years overall survival (OS) from primary tumor eradication and LT rises up to 53% [11].

The delayed diagnosis is not unusual, since these tumors are often discovered once patients developed metastasis. The liver represents one of the most common NETs metastatic sites, concerning more than 40% of patients, and is usually one of the most common causes of death [8,12]. Several treatment options are available for NETs liver metastases based on the number, pattern and pathology of metastatic lesions. The treatment options are namely liver resection and locoregional or systemic therapies. However, nowadays, the best management of hepatic NETs metastasis is still under debate: isolated lesions are still eligible for surgical resection, despite being burdened by high risks of recurrence and reduced chances to achieve an R0 resection, while subjects

affected by multiple hepatic lesions (>80% of total patients affected by hepatic NET metastasis) are often excluded from this option [13,14]. In fact, less than one-third of patients with NETs liver metastases are eligible for resection with curative intent. The palliative treatments, such as chemo or immunotherapy, ablative techniques or radionuclide therapies, are considered in extended diseases in order to relieve symptoms and reduce the detrimental effects of excessive serotonin release [8]. However, these therapies do not provide a curative chance, but only locoregional benefits in the early stages of treatment.

*3.2. Liver Transplantation: Outcomes*

LT was described as one of the viable options in unresectable metastatic NETs (constituting 0.2-0.3% of all LTs) with no other organs involvement [7,15]. The 5-year OS and disease-free survival after LT for NET ranges between 47% to 71%, and 20-32% respectively [7].

Unfortunately, patients with NETs are more prone to developing long-term recurrences than patients with HCC [7]. For this reason, multiple and specific prognostic factors for NETs need to be early identified to prevent negative outcomes and to provide an acceptable transplant benefit. The histologic grading system elaborated by the World Health Organization (WHO) divided well-differentiated NETs (G1-G2) according to the mitotic count. The G3 NETs (mitotic count > 20/HPF or Ki-67 > 20%) indicate poorly differentiated tumors with negative prognostic outcomes. The ELTR study demonstrated that the LT procedure is widely accepted when G1 and G2 NETs are diagnosed in the primary tumor since they guarantee better survival rates compared to G3 NETs (55% and 27% respectively) [10]. The extent of the involved hepatic parenchyma is another relevant prognostic factor: no more than 50% of the liver must be involved, since the tumor burden significantly affects the 5-year OS [10,16]. In order to analyze the biologic behavior and to not increase the surgical risks in performing two major surgeries at the same time, it is highly recommended to eradicate the primary tumor before LT [7,10].

Throughout the years, different groups tried to suggest different patient selection criteria. The Milan group adopted the aforementioned prognostic factors as LT selection criteria (i.e., histologic grade, portal drainage of the primary tumor, pre-transplant curative resection of all extrahepatic lesions, hepatic invasion less than 50%, duration of stable disease over six months), reporting encouraging outcomes (5 years patients and disease-free survival of 97% and 89%, respectively) [16,17]. However, although Milan criteria are currently the most widely accepted criteria, there is still no unanimous agreement. Moreover, as already highlighted in remarkable studies (Table 1), the importance of promising parameters such as patient age, waiting time for disease stabilization, primary tumor resection before LT, tumor bulk and <50% of hepatic tumor involvement still need further validation [10,18]. The study conducted by Mazzaferro et al. further proved that LT is strongly associated with positive survival and benefit outcomes. In the LT and no-LT group, the 10-year patient survival was 88.8% vs. 22.4%, while the median overall survival and recurrence-free survival were 62 vs. 20 months, respectively [18]. A prognostic factor influencing the survival benefit was represented by the primary tumor site and node (N) stage. In fact, greater benefits were achieved in those patients with no N tumor involvement (N0) or with primary colon/stomach NETs [10,18]. Likewise, in their remarkable studies, Le Treut et al. and Mazzaferro et al. identified different prognostic factors that widened the horizons of LT due to metastatic liver NETs [10,16]. In fact, more accurate selection criteria over the years were capable of extending the overall survival after LT for secondary liver tumors.

Table 1. Liver transplantation for neuroendocrine tumors (NETs).

| First Author (Year of Publication) | Study Design | Primary Tumor | N. of LT Patients | Tumor Location (%) | Patients with Metastases (N) | CHT preTx (N) | 5-Year OS (%) | 5 Year DFS (%) |
|---|---|---|---|---|---|---|---|---|
| Le Treut (2008) [19] | Multicenter experience | NET | 85 | Bronchial tree (5); Stomach (3); Jejunum (6); Ileum (16); Rectum (4); Duodenum, pancreas (41); Undetected (10) | 12 | 70 | 47 | 20 |
| Nguyen (2011) [20] | Registry databases | NET | 184 | N/A | N/A | N/A | 49.2 | NR |
| Gedaly (2011) [21] | Registry databases | NET | 150 | Metastatic carcinoid (51);Insulinoma (6); Glucagonoma (3); Gastrinoma (11); VIP tumour (9): Undetected (70) | 51 | N/A | 48 | 32 |
| Le Treut (2013) [10] | Registry databases | NET | 213 | Bronchial tree (16); Stomach (8); Jejunum (16); Ileum (48); Colon (5) Rectum (6); Duodenum, pancreas (97); Common bile duct (1); Undetected (17) | 56 | 161 | 52 | 30 |
| Sher (2015) [9] | Multicenter experience | NET | 85 | Duodenum, pancreas (42); Digestive tract (24) Undetected (19) | N/A | N/A | 52 | NR |
| Nobel (2015) [22] | Registry databases | NET | 120 | Carcinoid (34); ACTH-producing (1); Insulinoma (5); Glucagonoma (7); Gastrinoma (2); VIP tumour (4); Islet cell (9); Undetected (58) | N/A | N/A | 63 | NR |
| Pasqual (2016) [23] | Multicenter experience | NET | 4 | Stomach (4); Ileum (6); Colon (3); Duodenum (2); Esophagus (1); Pancreas (8); Lung (1); Undetected (1) | 22 | N/A | 50 | NR |
| Single multicenter experiences (1989–2020) [24–26] | Single centre experience | NET | 11 | Rectum (2); Small bowel (3); Pancreas (3); Stomach (1); Undetected (2) | 1 | 1 | 70.9 | 26.8 |

### 3.3. Colorectal Liver Metastasis—Natural History and Approach to Liver Transplantation

Colon-rectal cancer is one of the main oncological causes of death due to malignancy across the world and, unfortunately, approximately 20–25% of patients have already liver metastasis at referral [27]. Currently, hepatic resection is the gold standard treatment in resectable CRLM as it provides good results in terms of survival, even when negative prognostic factors are present [28].

Through a better understanding of the metastatic disease, innovative surgical procedures and the availability of modern therapies, the survival of patients with secondary liver tumors improved significantly during the last decades. Nowadays, LT for metastatic diseases is still based on the available data from clinical trials, hence there is still a long way ahead since these LT indications will be worldwide unanimously well-accepted as the gold standard of care. Currently, LT seems to be a compelling therapeutic option within strict selection criteria. Thanks to these criteria, results achieved so far both in NETs and CRLM seem to be encouraging.

In this regard, plenty of efforts were made in recent years to develop new surgical techniques to achieve R0 even in bilobar disease. In fact, in order to expand the hepatic resection criteria of bilateral CRLM, the two-stage hepatectomies were developed and it has been demonstrated that good results can be achieved [29]. However, approximately two-thirds of these patients are deemed unresectable during the diagnostic work-up and chemotherapy represents the only palliative option to improve the prognosis. Theoretically, LT could provide R0 margins as the procedure itself involves the removal of the whole liver.

### 3.4. Liver Transplantation: Outcomes

In view of this concept, in the early 90s LT was attempted but, subsequently, abandoned due to poor overall outcomes [4]. Therefore, the CRLM was an absolute contraindication for LT until 2013 when a milestone prospective pilot study (SECA-I) from the Norway group was published [30]. The study enrolled 21 patients with unresectable CRLM with no extrahepatic disease who previously underwent primary colonic tumor resection and had at least 6 weeks of chemotherapy. Afterward, all patients underwent diagnostic laparoscopy with hilar nodes sampling and, if negative, LT was subsequently performed. None of these patients received post LT chemotherapy. Even if the 1-year DFS was only 35%, the 1- 3- and 5-year OS of 95%, 68% and 60%, respectively were an extraordinary challenge [30]. Considering that chemotherapy alone has a reported 5-year overall survival of less than 10% [31], the results of SECA-I were highly remarkable even when compared to the survival reported for patients with resectable CRLM who underwent surgery [32]. Although the high recurrence rates and a short DFS, the authors stated that recurrences were mainly pulmonary and often solitary. Hence, these features enabled curative treatments and explained the excellent OS. Moreover, the authors were able to identify some risk factors (OSLO score) which were associated with poor outcomes: (i) largest tumor diameter measuring > 5.5 cm (ii) CEA levels > 80 mg/L (iii) less than two years from primary colonic cancer resection to LT and (iv) progressive disease on pre-transplant chemotherapy. Notably, almost 20% of patients in the SECA-1 experienced arterial complications and 50% of these patients lost their grafts due to hepatic artery thrombosis.

Four years later, a retrospective multicenter study from the Compagnons Hépato-Bilaires group confirmed these positive results [33]. Between 1995 and 2015, the LT was performed in 12 recipients with CRLM across different centers (Paris, Lisbon, Coimbra and Geneva) and, even if there was a heterogenous population, the authors reported a 5-year OS and DFS of 50% and 38%, respectively. Compared to the SECA-I study, half of the LT was performed as a planned strategy, while the other half of the patients were considered as having received "compassioned" LT. The selection between planned and compassioned LT was by no means standardized. This had an impact on the recurrence rate as patients who underwent planned transplantation following liver resection had lower recurrence rates when compared to the other group. In contrast to the SECA-I trial, no vascular complications were reported in this study.

In the more recent prospective SECA-II trial, 15 patients underwent LT for unresectable CRLM. Even if the DFS was the price to pay (i.e., the 1, 2, and 3 years DFS were 53%, 44%, and 35%, respectively), the OS achieved was of paramount importance (i.e., the 1, 3, and 5 years OS were 100%, 83%, and 83%, respectively) [34]. All the patients received pre-transplant systemic chemotherapy and at least 10% response was necessary in order to be included in the SECA-II protocol. Similar to the SECA-I study, none of the patients underwent adjuvant chemotherapy after LT. The immunosuppression (IS) protocol consisted of induction therapy (i.e., basiliximab), steroids, mycophenolate mofetil and tacrolimus for the first 4 to 6 weeks and mTOR inhibitor-based IS with tacrolimus and steroids withdrawal thereafter. The recurrences were mainly pulmonary and amenable for curative intent treatments, such as pulmonary resection. Remarkably, four patients had no relapse at the last available follow-up. Interestingly, the SECA-II cohort had significantly lower CEA levels and OSLO score as well as lower numbers of metastatic lesions and size of largest liver lesions than SECA-I study. In addition, one-third of the recipients did not experience any recurrence after 3 years, whereas in the SECA-I most of the patients had recurrences within 2 years. The more restrictive criteria of recipient selection could explain the differences between the two studies in terms of OS and DFS. However, it is important to mention that the burden of the disease in SECA-II was by far low: (i) more than 70% had pathological (y)pT3 tumor, (ii) more than 50% had received neoadjuvant treatment before primary colon cancer resection and (iii) 14/15 (93.3%) patients had the synchronous disease. In terms of surgical postoperative outcomes, in opposition to the SECA-I study, no major arterial complications were reported. The reports are summarized in Table 2.

The growing interest of LT in CRLM settings led to the development of several other procedures, such as (i) resection and partial liver segment 2/3 LT with delayed hepatectomy (RAPID) [35], (ii) living donor auxiliary partial LT and two-stage hepatectomy (LD-RAPID) [36,37], (iii) heterotopic transplantation of liver segments 2 and 3 using the splenic vein and artery after splenectomy and with delayed total hepatectomy (RAVAS) [38,39]. These new techniques have opened a new era in transplant oncology, also breaking the concept of living-donor LT for metastatic liver disease. However, outcomes are still under investigation and trials are ongoing in Europe and Canada, both with deceased and living donors: COLT (NCT03803436), LiverT(w)oHeal (NCT03488953), TRANSMET (NCT02597348), SECA-III (NCT03494946) and Toronto Living Donor study (NCT02864485) (Table 3). The Italian COLT study was designed as multicenter, non-randomized, open-label, controlled, prospective, parallel trial to assess OS of LT in CRLM versus chemotherapy in a matched cohort of patients who share the same baseline tumor characteristics. Likewise, the multicentric randomized open French trial TRANSMET aims to evaluate the 5-years OS of patients with CRLM treated with chemotherapy followed by LT or chemotherapy alone. The SECA-III trial will compare LT alone versus chemotherapy or other locoregional treatments, such as radiofrequency ablation, selective internal radiation therapy and trans-arterial chemoembolization. The German LiverT(w)oHeal and the Toronto Living Donor Study will focus on LT living donation for the treatment of CRLM. Interestingly, the German protocol was based on the concept of LD-RAPID and the aim is to analyze two-stage hepatectomy combined left-lateral living donor LT in patients with CRLM, whereas the Canadian protocol will be including living donor LT alone. Currently, no data are available.

**Table 2.** Liver transplantation for and colorectal liver metastasis (CRLM).

| First Author (Year of Publication) | Study Design | No. of LT Patients | Tumor Location | Patients with Metastases | Max Size at LT in cm (pt) | Chemothreapy preTx | 5-Year OS (%) | 5 Year DFS (%) |
|---|---|---|---|---|---|---|---|---|
| Hagness (2013) [30] | Prospective study | 21 | Colon 11 (52) Rectum 10 (48) | <6 (4) 6–9 (9) ≥10 (8) | <5 (12) 5–10 (5) >10 (4) | 21 | 60 | NR |
| Compagnons Hépato-Bilaires group (2017) [33] | Retrospective multicenter study | 12 | Colon 11 (91) Rectum 1 (9) | <6 (6) 6–9 (1) ≥10 (5) | <5 (4) 5–10 (1) >10 (7) | 11 | 50 | 38 |
| Dueland (2020) [34] | Prospective study | 15 | Colon 3 (20) Sigmoid 8 (53) Rectum 4 (27) | <6 (9) 6–9 (2) ≥10 (4) | <5 (15) 5–10 (0) >10 (0) | 15 | 83 | NR |

**Table 3.** Ongoing Clinical Trials on Liver Transplantation for Colorectal Liver Metastasis.

| Study Name | Year | Country | Indication | Estimated Patient Enrollment | Allocation | Type of Donor | Clinical Trial Number |
|---|---|---|---|---|---|---|---|
| COLT | 2019 | Italy | CRLM | 22 | Non-randomized | Cadaveric | NCT03803436 |
| LiverT(w)oHeal | 2018 | Germany | CRLM | 40 | NA | Living | NCT03488953 |
| Toronto Living Donor study | 2016 | Canada | CRLM | 20 | NA | Living | NCT02864485 |
| SECA-III | 2016 | Norway | CRLM | 30 | Randomized | Cadaveric | NCT03494946 |
| TRANSMET | 2015 | France | CRLM | 90 | Randomized | Cadaveric | NCT02597348 |

## 4. Strategies for Organs Allocation and Ethical Considerations

Different exchanging programs of donor livers in urgent cases such as Scandiatransplant provide favorable situations, such as the one related to the high organ donation rate in Norway. This is an important point that must be taken into account when evaluating these results, as a longer waiting time could have a major impact in dropping out from the LT list due to the progression of the disease. In countries with a long LT waiting time, such as the UK where the median time is 152 days [40], it could be difficult to add a new transplantable disease, as this, would inherently cause longer waiting time or even drop-out for other patients. In addition, the severe dearth of organs and mortality in the waiting list represent further problems. In the UK, the burden of waiting list mortality raises major ethical considerations, since it has been reported that the number of patients who die on the waiting list relative to the total number of LT is about 16% [40]. In this regard, recipients with metastatic liver disease usually have normal liver function. Thus, a MELD-based allocation system could not reflect the real burden of the disease, leaving these patients on the waiting list for longer periods with the risk of progressive disease which can, in the end, contraindicate the LT. On top of this, many countries have a high mortality rate on the waiting list due to a shortage of organs. Adding a new indication for LT which is currently not based on MELD and with higher numbers of patients who are likely to be eligible, would potentially worsen the burden of waiting list mortality.

On the other hand, these patients, that usually preserve liver function, do not demand optimal organs, thus allowing the utilization of extended criteria grafts that might be otherwise not suitable for other recipients. In addition, living donation has shown to be feasible for these recipients and it could offer some inherent advantages, such as optimizing the LT timing within chemotherapy regimens and increasing organ availability. The current differences among diverse organ allocating systems may appear sometimes controversial. Hence, ethical considerations are raised, since patients who might represent an urgency for LT in a specific allocation system and in a specific country, might not be considered as in need as in another country. Therefore, a deeper revision of these procedures may be needed to optimize future organ allocating organizations in order to not create any discrepancy between the patients on the waiting list.

## 5. Immunosuppression Administration

After LT, the m-TOR inhibitors, namely sirolimus and everolimus, have been preferred over calcineurin inhibitor (CNI)-based regimen, such as tacrolimus and cyclosporin, in both in the CLRM and NETs patients, for their potential anti-tumor effect [41–43]. Likewise, other evidence does not discourage the usage of CNIs, but supports that early CNIs minimization should be preferred in HCC recipients as it could decrease tumor recurrence [44]. Undoubtedly, IS plays a crucial role in tumor behavior and its accurate management is paramount to equipoise the risk of rejection and tumor recurrence. In this regard, recent data have demonstrated the feasibility of tapering IS for LT [45,46] and its complete weaning could be a valuable option even for cancer recipients [47]. Currently, all these data are mainly available for primary liver cancer, such as HCC, and the interaction between IS and secondary liver cancer has not been fully investigated yet. Nevertheless, no comparison with standard CNI-based IS is currently available for LT in NETs and CRLM, making it difficult to establish an adequate regimen for this cohort of recipients. As more evidence emerges in this field, tailoring the IS for NETs and CRLM recipients will become an interesting challenge for transplant physicians.

## 6. Conclusions

In the last few decades, additional knowledge on transplant oncology has improved the outcome of patients affected by NETs and CLRM. Certainly, the results of the ongoing prospective trials will assess the unanswered questions of the last trials and will better delineate the future role of LT in oncological patients.

Even though the results are highly encouraging, currently no guidelines support the routine use of LT for metastatic liver cancer. However, the latest evidence highlighted in the present manuscript confirms that the era of transplant oncology has clearly begun, and it is now a reality. Based on the aforementioned outcomes, it is expected that the ongoing trials, such as COLT, TRANSMET and SECA-III, will demonstrate the superiority of LT over the current standard treatments for metastatic liver tumor, paving the way for considering metastatic liver cancer in well-selected patients a new indication for LT. Living donation alone or combined with two-stage hepatectomy could be other feasible options for these recipients and results from the ongoing trial are awaited. As in the HCC setting [48,49], the optimization of patients' selection will be essential to ensure transplant benefit and achieve adequate outcomes in terms of both patient's OS and DFS. A multi-disciplinary transplant oncology team with specialized surgeons, oncologists, hepatologists and radiologists would be essential to guarantee the correct work-up and risk stratification for these patients. Finally, the allocation systems in different countries and the ethical considerations will need to be revised and adapted to the new upcoming transplant oncology scenario.

**Author Contributions:** T.M.M. and R.A. designed and edit the review; A.P. and C.G. wrote the paper; G.T. approved the final version. All authors have read and agreed to the published version of the manuscript.

**Funding:** This research received no external funding.

**Conflicts of Interest:** The authors declare no conflict of interest.

## Abbreviations

| | |
|---|---|
| ALF | Acute Liver Failure |
| CNI | Calcineurin Inhibitors |
| CRLM | Colorectal Liver Metastasis |
| ELTR | European Liver Transplant Registry |
| ESLD | End-Stage Liver Disease |
| HCC | Hepatocellular carcinoma |
| LT | Liver Transplantation |
| MELD | Model for End-stage Liver Disease |
| NET | Neuroendocrine tumors |
| PNET | Pancreatic neuroendocrine tumors |

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
