# Peer review of "The Revolution in Indication for Liver Transplantation: Will Liver Metastatic Disease Overcome the End-Stage Liver Disease in the Next Future?"

_2673-3943, doi:10.3390/transplantology1020011_

Round 1

Reviewer 1 Report

This is a quite ordinary review regarding liver transplantation for NET liver metastasis and CRLM.

Since the metastatic liver diseases are due to dissemination to the blood stream, liver transplantation should basically be contraindicated. In HCC patients, those with macroscopic vascular invasion are excluded from the indication for liver transplantation.

Liver transplantation for metastatic disease is still based  on clinical trail, and far from well-accepted clinical practice. Authors had better take this point into account, and qualify the pro expressions for this procedure.

Author Response

Thank you for your kind comments, this is a really interesting point. In the present manuscript we described only the “state of the art” regarding liver transplantation for metastatic disease.However, this point of view concerning the topic did not emerge sufficiently. Therefore, further modifications were acknowledged in the present paper and your comments addressed.

Reviewer 2 Report

This is a review article on the liver transplantation for the neuroendocrine tumors and colorectal liver metastases.

I have some comments.

  1. (P3, L117) Make a new line from “Unfortunately, patients with NETs …”.
  2. (P6, L243~245) Delete the sentences “The Table 1.. in transplant community.” Make a new line from “It is worth to …”.
  3. (P6, L262) Make a new line from “On the other hand …”.
  4. (Table 1) Divide the data into two, the one for liver transplantation for NET and the other for liver transplantation for CRLM. Each table should be placed after the text “Neuroendocrine tumor” and “Colorectal liver metastasis”, respectively.

Author Response

Thank you for your kind remarks.

Your suggestion has been addressed in the manuscript.

Reviewer 3 Report

Dr Manzia and colleagues summarise the literature to date regarding liver transplantation in the setting of metastatic neuroendocrine tumour and colorectal liver metastasis. The review topic is important and the authors cover relevant aspects, however, I have the following suggestions:

  1. The article is written in an odd format.  Although there is a methodology section, the results are not presented as a formal systematic review nor are the headings and flow of the manuscript reflective of a narrative review.  I would suggest removing the "discussion" heading and incorporating the text into the NET and CRLM sections.  Each of these should have subsections (eg epidemiology/natural history, approach to LT and outcomes) to summarise the evidence, then the section on immunosuppression augmentation and then a final concluding paragraph.  
  2. A figure/algorithm with an approach to both NET and CRLM in the setting of LT would be useful (eg patient selection, projected outcomes and post-LT considerations)
  3. The ethical considerations should be discussed in more detail. The comprehensive discussion about different allocation systems is appreciated.
  4. Tumour characteristics in Table 1 would be helpful
  5. A table to summarise ongoing clinical trials would also be helpful
  6. The English language requires careful revision.  There are several grammatical errors (eg. "next future" "recommend the primary tumour removal", "are prone on" etc etc.
  7. There are minor typographical errors that also need to be addressed

Author Response

Thank you for your suggestions.

We addressed most of concerns, the grammar mistakes as well as English language have been revised throughout the text.

The available tumors characteristics were stated in the tables.

Ethical considerations and allocation systems were further discussed

A table summarizing ongoing clinical trials was provided in the manuscript with available details.

we consciously decide to not provide an algorithm at this step since there is not a worldwide consensus, scientific recommendation or guidelines on this topic (either on CLRM and NETs). The risk could be the scientific confusion on the readers side.